# Clonal hematopoiesis is associated with risk of severe Covid-19

Kelly L. Bolton[1,21 ✉], Youngil Koh [2,3,4,21], Michael B. Foote [5,21], Hogune Im [3,21], Justin Jee [5,21], Choong Hyun Sun[3,21], Anton Safonov[5,21], Ryan Ptashkin[6], Joon Ho Moon[7], Ji Yeon Lee [8], Jongtak Jung [9], Chang Kyung Kang [2], Kyoung-Ho Song [9], Pyoeng Gyun Choe [2], Wan Beom Park [2], Hong Bin Kim [9], Myoung-don Oh [2], Han Song [3], Sugyeong Kim[3], Minal Patel[10], Andriy Derkach [11], Erika Gedvilaite [6], Kaitlyn A. Tkachuk[5], Brian J. Wiley[1], Ireaneus C. Chan[1], Lior Z. Braunstein[12], Teng Gao [13], Elli Papaemmanuil [13], N. Esther Babady[5,14], Melissa S. Pessin [14], Mini Kamboj [5], Luis A. Diaz Jr[5], Marc Ladanyi[6], Michael J. Rauh[15], Pradeep Natarajan [16,17], Mitchell J. Machiela [18], Philip Awadalla [19], Vijai Joseph [20], Kenneth Offit[20], Larry Norton[5], Michael F. Berger [6,13], Ross L. Levine [5], Eu Suk Kim [9,21 ✉], Nam Joong Kim [2,21 ✉] & Ahmet Zehir [6,21 ✉]

Acquired somatic mutations in hematopoietic stem and progenitor cells (clonal hematopoiesis or CH) are associated with advanced age, increased risk of cardiovascular and malignant diseases, and decreased overall survival. These adverse sequelae may be mediated by altered inflammatory profiles observed in patients with CH. A pro-inflammatory immunologic profile is also associated with worse outcomes of certain infections, including SARS-CoV-2 and its associated disease Covid-19. Whether CH predisposes to severe Covid-19 or other infections is unknown. Among 525 individuals with Covid-19 from Memorial Sloan Kettering (MSK) and the Korean Clonal Hematopoiesis (KoCH) consortia, we show that CH is associated with severe Covid-19 outcomes (OR = 1.85, 95% = 1.15–2.99, p = 0.01), in particular CH characterized by non-cancer driver mutations (OR = 2.01, 95% CI = 1.15–3.50, p = 0.01). We further explore the relationship between CH and risk of other infections in 14,211 solid tumor patients at MSK. CH is significantly associated with risk of *Clostridium Difficile* (HR = 2.01, 95% CI: 1.22–3.30, p = 6×10$^{-3}$) and *Streptococcus/Enterococcus* infections (HR = 1.56, 95% CI = 1.15–2.13, p = 5×10$^{-3}$). These findings suggest a relationship between CH and risk of severe infections that warrants further investigation.

A full list of author affiliations appears at the end of the paper.

Acquired mutations that lead to clonal expansion are common in the normal aging hematopoietic system (clonal hematopoiesis, or CH), yet are known to alter stem/progenitor and lymphoid function and response to environmental stressors, including systemic infections[1–4]. The mutational events that drive CH overlap with known drivers of hematologic malignancies. However, the majority of mutations in CH appear to occur outside of canonical cancer driver genes[5,6]. The impact of individual mutational events on hematopoietic stem and progenitor cells differs by the nature of the genomic aberration. For example, chromosomal aneuploidies result in a predisposition for lymphoid fate specification and transformation[7,8], whereas point mutations in *DNMT3A* result in increased myeloid differentiation[9]. Heterogeneity also exists across CH phenotypes by driver gene in regards to its impact on inflammatory signaling[2]. For example, mutations in *TET2* result in the heightened secretion of several cytokines including IL-1β/IL-6 signaling that may partially explain the increased risk of cardiovascular disease[1,3,10]. Moreover, systemic infections and the resultant inflammatory signals can lead to increased clonal fitness of *TET2* mutant cells and clonal expansion[4,11,12]. Despite these important insights, the relationship between different CH-associated mutations, infectious disease risk, and severity has not been studied. The severity of Covid-19 is also associated with advanced age, cardiovascular disease, and elevated circulating IL-6 levels; features that are also associated with CH[13–17]. Given the common inflammatory profile of CH and severe Covid-19 infection, we investigated the relationship between CH and Covid-19 disease severity. We went on to study the relationship between CH and other serious infections.

Here, we show that CH is associated with an increased risk of severe Covid-19 outcomes in a cohort of solid tumor patients and healthy individuals. Within solid tumor patients, CH is associated with an increased risk of *Clostridium Difficile* and *Streptococcus/Enterococcus* infections.

## Results

**Patient characteristics, CH, and Covid-19 assessment.** Our study included patients from two separate cohorts. The first cohort was composed of patients with solid tumors treated at Memorial Sloan Kettering Cancer Center (MSK) with blood previously sequenced using MSK-IMPACT, a previously validated targeted gene panel capturing all commonly mutated CH-associated genes (Supplementary Data 1)[18]. Of these patients, 1636 were tested for SARS-CoV-2 (the virus that causes Covid-19) RNA between 1st March 2020 and 1st July 2020; 413 (25%) individuals tested positive for SARS-CoV-2 (Methods and Table 1). The second cohort included 112 previously healthy individuals without cancer who were hospitalized for Covid-19 between January and April 2020 at four tertiary hospitals in South Korea (KoCH cohort). The KoCH cohort was sequenced using a custom targeted NGS panel from Agilent (89 genes), which was designed to include commonly occurring CH genes (Supplementary Data 2).

For both cohorts, the primary outcome was severe Covid-19 infection, defined as the presence of hypoxia requiring >1 L supplemental oxygen with documented hypoxia (oxygen saturation <94%). To determine the association between severe Covid-19 and CH separately in each cohort, we used multivariable logistic regression adjusting for covariates including age, gender, smoking, and prior Covid-19-related comorbidities including cardiovascular disease, hypertension, and chronic obstructive pulmonary disease (COPD)/asthma. The MSK cohort was also adjusted for cancer primary site and exposure to cytotoxic cancer therapy before and after blood draw. We then performed a fixed-effects meta-analysis to estimate the association in the overall population. The full statistical rationale is further described in the Methods section.

**Association of CH with severe Covid-19 infections.** Among Covid-19-positive individuals, 23% ($N = 94$) and 61% ($N = 68$) had severe disease in the MSK and KoCH cohorts, respectively (Table 1, Supplementary Fig. 1). Overall, CH was observed in 35% of Covid-19-positive cases at MSK and 21% in KoCH. Of note, when restricting the MSK-IMPACT panel to the 89 genes included in the KoCH panel, 20% of Covid-19-positive cases at MSK had CH. In the MSK cohort, CH was observed in 51% and 30% of patients with severe versus non-severe Covid-19, respectively (adjusted OR: 1.85, 95% CI 1.10–3.12, $p = 0.02$) (Fig. 1). A sensitivity analysis of the MSK data limited to hospitalized patients ($N = 117$) yielded similar, albeit non-significant, results (OR = 1.6; 95% CI = 0.5–5.2). In the KoCH cohort, CH was observed in 25% and 16% of patients with severe versus non-severe Covid-19, respectively, (adjusted OR 1.85, 95% CI 0.53–6.43, $p = 0.33$) (Fig. 1). In a fixed-effects meta-analysis of odds ratio estimates from the multivariable logistic regression models employed in each separate cohort analysis, the presence of CH was associated with an increased risk of severe Covid-19 (OR = 1.85, 95% = 1.15–2.99, $p = 0.01$) (Fig. 1). We did not see evidence of heterogeneity in the strength of the association in the MSK and KoCH cohorts. This is in line with evidence of similarity in predictors of Covid-19 severity in cancer and non-cancer populations[13,19–21] but requires further examination in larger cohorts. The odds ratio we observed between CH and severe Covid-19 we observed is similar to that previously reported by Hameister et al.[22] (OR = 1.2; 95% CI = 0.5–3.0).

Using previously described methods[18], CH mutations were classified as known or hypothesized cancer putative drivers (PD-CH) or non-putative drivers (non-PD-CH). In order to explore the association between particular mutation types and Covid-19 severity, we performed a stratified analysis of Covid-19 severity by PD-CH versus non-PD-CH status using multivariate logistic regression including the covariates from the main model. A significant association was observed between non-PD-CH and severe Covid-19 (OR = 2.01, 95% CI = 1.15–3.50, $p = 0.01$), as well as between silent (synonymous) CH and severe Covid-19 (OR = 2.58, 95% CI 1.01–6.61, $p = 0.05$) (Supplementary Fig. 2). There was not a statistically significant association between PD-CH and severe Covid-19 infection (OR = 1.15, 95% CI = 0.61–2.02, $p = 0.62$) (Supplementary Fig. 2). Most non-PD mutations in severe Covid-19 cases occurred in non-recurrently mutated genes (65% at MSK and 77% in KoCH, Supplementary Fig. 3). In additional exploratory analyses, we characterized the association between CH variant allele frequency (VAF), mutation number, and severe Covid-19. The strength of the association between CH and severe Covid-19 was similar among patients with one CH mutation (OR = 1.78, 95% CI = 1.02–3.09, $p = 0.04$) and multiple CH mutations (OR = 1.97, 95% CI = 1.03–3.78, $p = 0.04$) (Supplementary Fig. 4). Patients with a maximum CH VAF of >5% showed a significant association with severe Covid-19 (OR = 1.89, 95% CI = 1.04–3.43, $p = 0.04$, Supplementary Fig. 5). In an exploratory analysis, we also observed a significant association between the risk of invasive/non-invasive ventilation and non-driver CH (OR = 2.18, 95% CI 1.08–4.40, $p = 0.03$: Supplementary Fig. 6). These data suggest that the presence of CH and resultant alterations in hematopoietic differentiation, and not specific mutant alleles, is predictive of Covid-19 disease severity.

To study whether exposure to chemotherapy might modify the relationship between non-PD-CH, PD-CH, and Covid-19

**Table 1 Characteristics of study participants.**

| | MSK | | | | KoCH | |
|---|---|---|---|---|---|---|
| | Severe covid (N = 94) | Non-severe covid (N = 319) | Negative (N = 1223) | Untested (N = 7681) | Severe covid (N = 68) | Non-severe covid (N = 44) |
| *Age(y)* | | | | | | |
| 0–50 | 12 (12.8%) | 84 (26.3%) | 308 (25.2%) | 1712 (22.3%) | 6 (8.8%) | 16 (36.4%) |
| 50–60 | 17 (18.1%) | 93 (29.2%) | 303 (24.8%) | 1741 (22.7%) | 10 (14.7%) | 6 (13.6%) |
| 60–70 | 32 (34.0%) | 83 (26.0%) | 356 (29.1%) | 2358 (30.7%) | 23 (33.8%) | 8 (18.2%) |
| 70–80 | 28 (29.8%) | 46 (14.4%) | 211 (17.3%) | 1523 (19.8%) | 15 (22.1%) | 9 (20.5%) |
| 80+ | 5 (5.3%) | 13 (4.1%) | 45 (3.7%) | 347 (4.5%) | 14 (20.6%) | 5 (11.4%) |
| *Gender* | | | | | | |
| Female | 54 (57.4%) | 169 (53.0%) | 675 (55.2%) | 4569 (59.5%) | 28 (41.2%) | 23 (52.3%) |
| Male | 40 (42.6%) | 150 (47.0%) | 548 (44.8%) | 3112 (40.5%) | 40 (58.8%) | 21 (47.7%) |
| *Smoking* | | | | | | |
| Never | 38 (40.4%) | 157 (49.2%) | 565 (46.2%) | 3687 (48.0%) | 34 (50.0%) | 30 (68.2%) |
| Current/former | 55 (58.5%) | 154 (48.3%) | 639 (52.2%) | 3910 (50.9%) | 10 (14.7%) | 13 (29.5%) |
| Missing | 1 (1.1%) | 8 (2.5%) | 19 (1.6%) | 84 (1.1%) | 24 (35.3%) | 1 (2.3%) |
| *Hypertension* | | | | | | |
| No | 38 (40.4%) | 183 (57.4%) | 673 (55.0%) | 4422 (57.6%) | 34 (50.0%) | 26 (59.1%) |
| Yes | 56 (59.6%) | 136 (42.6%) | 550 (45.0%) | 3259 (42.4%) | 34 (50.0%) | 18 (40.9%) |
| *Coronary artery disease* | | | | | | |
| No | 80 (85.1%) | 285 (89.3%) | 1104 (90.3%) | 7058 (91.9%) | 63 (92.6%) | 40 (90.9%) |
| Yes | 14 (14.9%) | 34 (10.7%) | 119 (9.7%) | 623 (8.1%) | 5 (7.4%) | 4 (9.1%) |
| *COPD/asthma* | | | | | | |
| No | 78 (83.0%) | 271 (85.0%) | 1030 (84.2%) | 6531 (85.0%) | 64 (94.1%) | 41 (93.2%) |
| Yes | 16 (17.0%) | 48 (15.0%) | 193 (15.8%) | 1150 (15.0%) | 4 (5.9%) | 3 (6.8%) |
| *Diabetes* | | | | | | |
| No | 72 (76.6%) | 256 (80.3%) | 1039 (85.0%) | 6681 (87.0%) | 48 (70.6%) | 36 (81.8%) |
| Yes | 22 (23.4%) | 63 (19.7%) | 184 (15.0%) | 1000 (13.0%) | 20 (29.4%) | 8 (18.2%) |
| *Race* | | | | | | |
| White | 54 (57.4%) | 199 (62.4%) | 912 (74.6%) | 5728 (74.6%) | 0 (0.0%) | 0 (0.0%) |
| Missing | 5 (5.3%) | 24 (7.5%) | 52 (4.3%) | 331 (4.3%) | 0 (0.0%) | 0 (0.0%) |
| Non-White | 35 (37.2%) | 96 (30.1%) | 259 (21.2%) | 1622 (21.1%) | 68 (100%) | 44 (100%) |
| *Month of Covid Dx* | | | | | | |
| January | 0 (0.0%) | 0 (0.0%) | 0 (0.0%) | | 2 (2.9%) | 6 (13.6%) |
| February | 0 (0.0%) | 0 (0.0%) | 0 (0.0%) | | 14 (20.6%) | 9 (20.5%) |
| April | 47 (50.0%) | 141 (44.2%) | 246 (20.1%) | | 1 (1.5%) | 5 (11.4%) |
| March | 44 (46.8%) | 93 (29.2%) | 89 (7.3%) | | 26 (38.2%) | 13 (29.5%) |
| May | 3 (3.2%) | 75 (23.5%) | 681 (55.7%) | | 5 (7.4%) | 8 (18.2%) |
| June | 0 (0%) | 10 (3.1%) | 206 (16.8%) | | 20 (29.4%) | 3 (6.8%) |
| Missing | 0 (0%) | 0 (0%) | 1 (0.1%) | | | |
| BMI (mean) | 27.5 | 28.2 | 26.5 | 27.8 | 23.5 | 23.7 |
| *Cytotoxic therapy prior to blood draw* | | | | | | |
| No | 59 (62.8%) | 199 (62.4%) | 742 (60.7%) | 5549 (72.2%) | | |
| Yes | 35 (37.2%) | 120 (37.6%) | 481 (39.3%) | 2132 (27.8%) | | |
| *Cytotoxic therapy after blood draw* | | | | | | |
| No | 30 (31.9%) | 130 (40.8%) | 349 (28.5%) | 4208 (54.8%) | | |
| Yes | 64 (68.1%) | 189 (59.2%) | 874 (71.5%) | 3473 (45.2%) | | |
| *Primary tumor site* | | | | | | |
| Other | 66 (70.2%) | 266 (83.4%) | 927 (75.8%) | 6097 (79.4%) | | |
| Thoracic | 28 (29.8%) | 53 (16.6%) | 296 (24.2%) | 1584 (20.6%) | | |

severity, we performed a series of stratified analyses by prior exposure to cytotoxic chemotherapy within the MSK cohort. We observed a similar enrichment of non-PD-CH among individuals with severe Covid-19 among those exposed to cytotoxic therapy (2.12, 95% CI 0.71–6.22, $p = 0.17$) and unexposed (1.88, 95% CI 0.98–3.90, $p = 0.09$). The relationship between non-PD-CH and Covid-19 severity did not appear to be greatly modified by exposure to cytotoxic therapy. Similar to our findings overall for CH-PD, among those not previously exposed to chemotherapy, no association was observed between CH-PD and Covid-19 severity (OR = 0.82, 95% CI 0.34–1.83, $p = 0.63$). We did, however, observe a non-significant trend towards a positive association between CH-PD and Covid-19 severity among those previously exposed to chemotherapy (OR = 2.4, 95% CI

0.86–6.74, $p = 0.09$). Of note, we also observed a trend towards a higher proportion of individuals with CH-PD mutations in *PPM1D* among those with severe Covid-19 (5%, $N = 5$) compared with non-severe Covid-19 (1%, $N = 2$) (Supplementary Fig. 7). This provides some evidence that CH-PD in cancer patients with severe Covid-19 may reflect poor hematopoietic reserve after chemotherapy and in turn worse immune response to infection. However, this would require further investigation in larger populations of cancer patients.

**Association between CH and risk of diverse infections.** Given the evidence of an association between CH and Covid-19 severity, we sought to explore the relationship between CH and other types of infections. We analyzed billing codes from 14,211 solid tumor

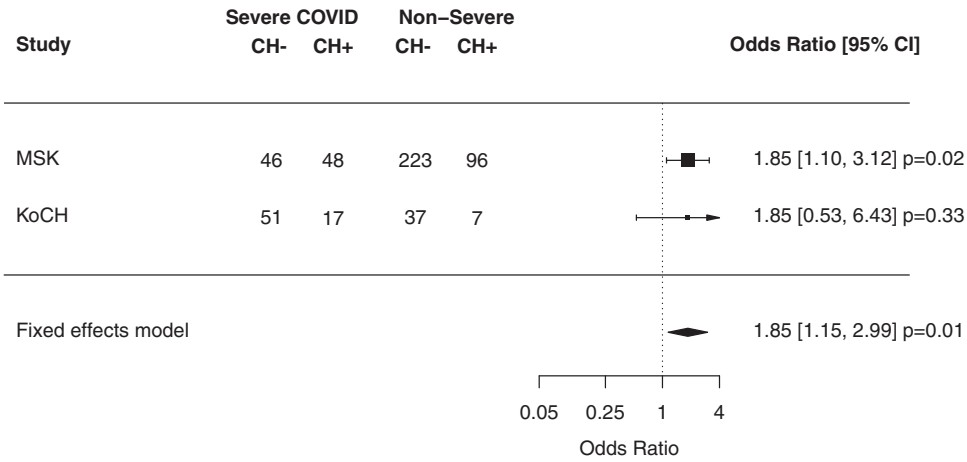

**Fig. 1 Association between CH and Covid-19 severity.** Shown are the results from logistic regression adjusted for age, gender, race, smoking, diabetes, cardiovascular disease, COPD/asthma, BMI, and month of Covid-19 diagnosis in 525 individuals. The MSK cohort was also adjusted for cancer primary site, exposure to cytotoxic cancer therapy before and after blood draw. Summary statistics for a fixed-effects meta-analysis are shown. Any CH includes both driver (CH-PD) and non-driver mutations (CH-non-PD).

patients treated at MSK who underwent blood sequencing by MSK-IMPACT. Using a previously established phenome-wide association study methodology[23], we mapped patient ICD-9 and ICD-10 billing codes to categories of infectious disease. Multivariable Cox proportional hazards regression was used to estimate the hazard ratio (HR) for risk of infection among CH-positive compared with CH-negative individuals. All models were adjusted for age, race, smoking, gender, cumulative to cytotoxic therapy prior to blood draw, cumulative exposure to cytotoxic therapy after blood draw, and primary tumor site. Given the number of model covariates, we limited the analysis to 32 infection subclasses that affected at least 80 individuals (see Methods). Multiple infection types were associated with CH, although many associations were not statistically significant after adjustment for multiple testing (Fig. 2A, Supplementary Data 3). CH was significantly (false-discovery rate (FDR)-corrected $p$ value < 0.10) associated with the onset of two infection subclasses: *C. difficile* infection (HR = 2.01, 95% CI: 1.22–3.30, $p = 6 \times 10^{-3}$) and *Streptococcus/Enterococcus* infection (HR = 1.56, 95% CI = 1.15–2.13, $p = 5 \times 10^{-3}$). When stratified by CH-mutation characteristics, patients with two or more CH mutations had a stronger association with *C. difficile* infection (OR = 3.37, 95% CI = 1.79–6.33, $p = 2 \times 10^{-4}$) compared with patients with one CH mutation (OR = 1.42, 95% CI = 0.75–2.67, $p = 0.28$). The association between CH and *C. difficile* infection was significant for mutations with a VAF of >5% (OR = 2.54, 95% CI = 1.39-4.63, $p = 0.002$) but not mutations with a VAF of 2–5% (OR = 1.59, 95% CI = 0.83–3.05, $p = 0.17$). Similar to Covid-19 severity, the association between CH and *C. difficile* infection was significant for non-PD-CH (OR = 2.11, 95% CI = 1.25–3.59, $p = 0.01$) and silent mutations (OR = 2.64, 95% CI = 1.20–5.80, $p = 0.02$) but not CH-PD (OR = 1.42, 95% CI = 0.74–2.82, $p = 0.39$) (Fig. 2B). Given the strong relationship between prior antibiotic use and *C. difficile* infection risk, we performed a sensitivity analysis including the number of courses of antibiotics received after MSK-IMPACT blood draw. There was no clear association between CH and antibiotic use and no effect of inclusion of this as a covariate in the model (Supplementary Fig. 8).

## Discussion

In summary, we show that CH is associated with increased Covid-19 severity in a heterogeneous, international cohort of cancer and non-cancer patients. In a large cancer patient cohort, CH is also associated with other severe infections, namely *Streptococcus/Enterococccus* and *C. difficile* infections. Future studies in large patient populations are needed to further characterize the association between CH and infection severity and whether this might differ in cancer compared with non-cancer populations and by other patient characteristics including race and age among others. Our exploratory analysis suggests that the relationship between CH and Covid-19 and CH and *C. difficile* infection may be partly driven by non-driver CH. Clonal expansions characterized by non-driver mutational events could be facilitated by multiple mechanisms. Many classes of genetic alterations, such as copy number events (CNVs), structural variants, non-coding, and epigenetic changes, are not detectable using the targeted panels included in this study. As such, the observed events that are highly enriched in CH could be "passenger" mutations that co-occur with a positively selected, undetected "driver" mutation such as recurrent CNVs[6,7]. Alternatively, driver mutations may have been incompletely classified as "non-driver" events using our methodology. However, cancer driver genes tend to recur in multiple patients, and the majority of witnessed non-driver mutated genes in our cohort were non-recurrent suggesting that clonal expansion, and not the specific event driving clonal expansion, may be associated with Covid-19 disease severity. Owing to the small sample sizes, the association between CH, infection risk, and Covid-19 severity, including differences by gene and driver mutation status, will need to be further studied in larger cohorts.

The hematopoietic system is a key regulator of inflammation and immunity. A substantial body of evidence now links somatic alterations in hematopoietic stem and progenitor cells to a variety of health outcomes, with inflammation emerging as a key mediator[2–5,10–13]. Our data, along with results in the manuscript by Zekavat et al.[24], demonstrate a similar association between CH, risk of severe Covid-19, and certain infection types. This association may be due to residual confounding by variables that are unknown and unaccounted for in our models. Even in this situation, CH as a biomarker of poor outcome could be a useful clinical predictor. Alternatively, this could be owing to CH-induced changes in the hematopoietic stem, progenitor, and lymphoid cell function impacting immune regulation and infection response. Shared and/or distinct processes may underlie the

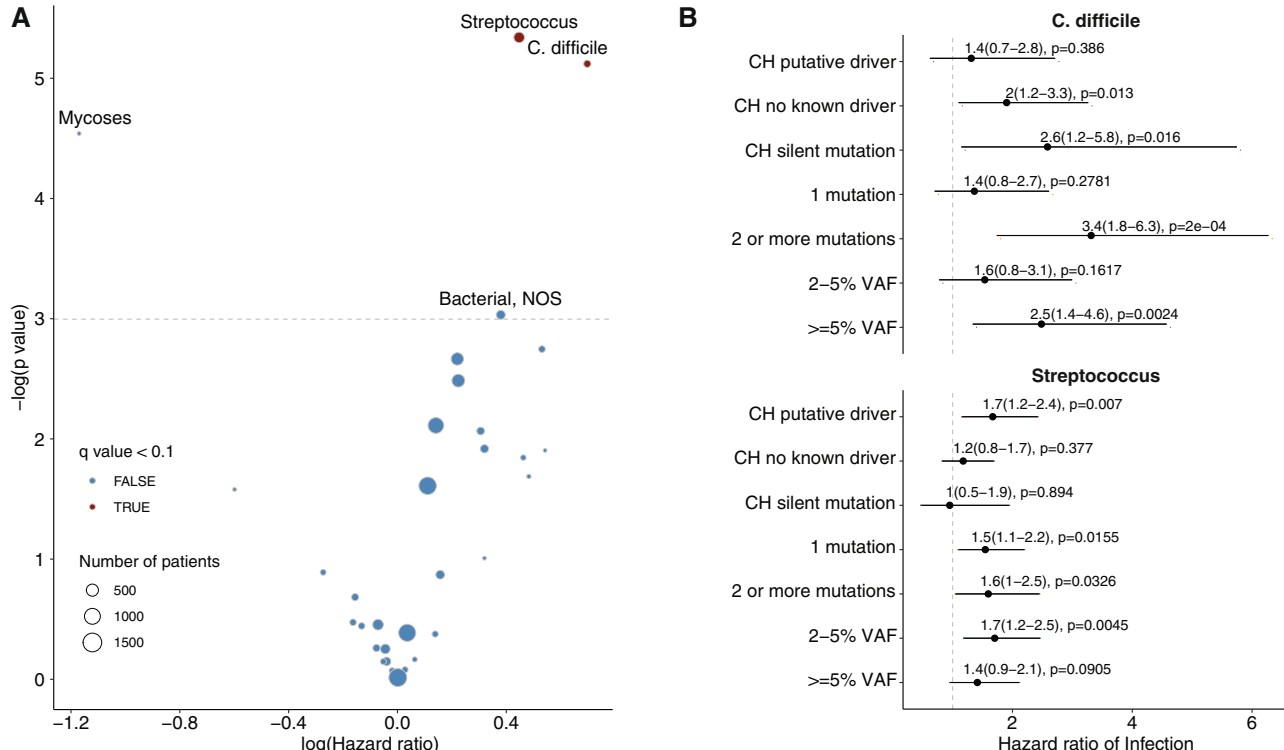

**Fig. 2 Association between CH and risk of infection in 14,211 solid tumor patients. A** Volcano plot of the log(Hazard ratio) of infection with CH using multivariable cox proportional hazards regression. Any CH includes both driver (CH-PD) and non-driver mutations (CH-non-PD). **B** Association between CH subtype defined by CH putative driver status and risk of *Clostridium Difficile* (N = 189) and *Streptococcus/Enterococcus* (N = 501) infection using cox proportional hazards regression. All models were adjusted for age, gender, race, smoking, cancer primary site, cumulative exposure to cytotoxic cancer therapy before and after blood draw. Shown is the hazard ratio for infection, associated 95% confidence interval, and *p* value (not adjusted or multiple comparisons).

associations we observed between CH, Covid-19 severity, and infection risk. Future investigation including functional studies will be important to clarify these mechanisms and to develop potential interventional strategies to attenuate inflammation, clonal expansion, and infectious sequelae in patients with and without cancer.

## Methods

### Sample ascertainment and clinical data extraction

*MSK cohort*. Subjects had a tumor and blood sample (as a matched control) sequenced using MSK-IMPACT on an institutional prospective tumor-sequencing protocol (ClinicalTrials.gov number, NCT01775072) before July 1st 2019. This study was approved by the MSKCC Institutional Review Board (IRB). A subset of patients who underwent tumor genomic profiling as the standard of care did not directly consent in which case an IRB waiver was obtained to allow for inclusion into this study. The study population for Covid-19 analyses included a subset of 9,307 patients with non-hematologic cancers who were alive on 1st March 2020. Subjects who had a hematologic malignancy diagnosed after MSK-IMPACT testing or who had an active hematologic malignancy at the time of blood draw were excluded. Demographics, smoking history, exposure to oncologic therapy, and primary tumor site were extracted from the electronic health record (EHR). The accuracy of populated information was manually checked in the EHR by three independent physicians (K.B., M.F, A.S.). The presence of co-existing medical comorbidities known to correlate with Covid-19 severity including diabetes, COPD, asthma, hypertension, and cardiovascular disease, were ascertained from ICD-9 and ICD-10-billing codes. SARS-CoV-2 status was determined using RT-PCR from laboratory records available through the MSK EHR. SARS-CoV-2 testing was performed both in the inpatient and outpatient settings. We defined severe Covid-19 as the presence of hypoxia requiring supplemental oxygen (supplemental oxygen device >1 L with oxygen saturation <94%) resulting from Covid-19 infection. Hypoxia was selected as our measure of disease severity due to the frequency of its occurrence and since it was based on an objective measure (i.e., oxygen saturation). There were seven subjects with Covid-19 for whom there was minimal documentation of clinical course following Covid-19 infection and these

individuals were excluded. There were three individuals with metastatic cancer and progression of disease at the time of Covid-19 where it was unclear whether documented hypoxia could be attributed to Covid-19 or disease progression. These subjects were also excluded. For future studies, we also provide individual-level data in regards to Covid-19 related mortality (see data availability for data access).

*KoCH cohort*. Laboratory-confirmed patients with Covid-19 who were hospitalized between January and April 2020 in four tertiary hospitals in the Republic of Korea were approached for consent to this study. Blood was drawn following confirmation of Covid-19 positivity. All four hospitals have been running national-designated isolation units, which are located in Seoul, Gyeonggi, or Daegu. These provinces had the highest numbers of Covid-19 cases during the period[14]. The indications for hospitalization in the KoCH cohort during this time period were complex and, in some situations, individuals who were asymptomatic or minimally symptomatic were hospitalized. Clinical and laboratory characteristics were retrospectively reviewed using the electronic medical record systems of each institution. Hypoxia requiring supplemental oxygen was defined as supplemental oxygen device >1 L with oxygen saturation <94%, resulting from Covid-19 infection. The Seoul National University Hospital IRB, Seoul National University Bundang Hospital IRB, National Medical Center IRB, and Kyungpook National University Hospital IRB approved the study (IRB numbers 2003-141-1110, B-2006/616-409, NMC-2008-050, KNUH-2020-04-069-001, respectively). Written informed consent for all participants was obtained per IRB recommendations. Subjects who had an active malignancy at the time of blood draw were excluded.

### Sequencing and variant calling

*MSK cohort*. MSK-IMPACT is an FDA-authorized hybridization capture-based next-generation sequencing assay encompassing all protein-coding exons from the canonical transcript of 341, 410, or 468 cancer-associated genes (Supplementary Data 1). MSK-IMPACT is validated and approved for clinical use by New York State Department of Health Clinical Laboratory Evaluation Program. The sequencing test utilizes genomic DNA extracted from formalin-fixed paraffin-embedded (FFPE) tumor tissue as well as matched patient blood samples. DNA is sheared and DNA fragments are captured using custom probes. MSK-IMPACT contains most of the commonly reported CH genes with the exception that earlier

versions of the panel did not contain *PPM1D* or *SRSF2* (5% of individuals were sequenced on this earlier version).

Pooled libraries were sequenced on an Illumina HiSeq 2500 with 2 × 100 bp paired-end reads. Sequencing reads were aligned to the human genome (hg19) using BWA (0.7.5a). Reads were realigned around indels using ABRA (0.92), followed by base-quality score recalibration with Genome Analysis Toolkit (GATK) (3.3-0). Median coverage in the blood samples was 497×, and median coverage in the tumors was 790×. Variant calling for each blood sample was performed unmatched, using a pooled control sample of DNA from 10 unrelated individuals as a comparator. Single-nucleotide variants (SNVs) were called using Mutect (1.1.4) and VarDict (1.4.6). Insertions and deletions were called using Somatic Indel Detector (2.3) and VarDict. Variants that were called by two callers were retained. Dinucleotide substitution variants were detected by VarDict and retained if any base overlapped an SNV called by Mutect. All called mutations were genotyped in the patient-matched tumor sample. Mutations were annotated with VEP (version 86) and OncoKb. We applied a series of post-processing filters to further remove false-positive variants caused by sequencing artifacts and putative germline polymorphisms as previously described[18] and as detailed in the Supplemental Methods section.

*KoCH cohort.* Blood-derived DNA was sequenced using a custom panel of 89 genes frequently mutated in CH. All NGS libraries were prepared using the Agilent SureSelect XT HS and XT Low input enzymatic fragmentation kit. Pooled Libraries were sequenced on an Illumina NovaSeq6000 with 2 × 150 bp paired-end reads. Sequencing reads were trimmed with SeqPrep (v0.3) and Sickle (v1.33) and aligned to the human genome (hg19) using BWA-MEM (v0.7.10). PICARD(v1.94) was used for duplicate marking followed by indel realignment and base-quality score recalibration with GATK light(v2.3.9). The mean depth of coverage of samples was higher than 800×. Variant calling was performed using SNver(v0.4.1), LoFreq(v0.6.1), GATK UnifiedGenotyper(v2.3.9) for SNVs. For Insertions and deletions, in-house caller was used[25].

**Variant annotation**. Variants from the MSK and KoCH cohort were uniformly annotated according to evidence for functional relevance in cancer (putative driver or CH-PD). We annotated variants as oncogenic if they fulfilled any of the following criteria: (1) truncating variants in *NF1, DNMT3A, TET2, IKZF1, RAD21, WT1, KMT2D, SH2B3, TP53, CEBPA, ASXL1, RUNX1, BCOR, KDM6A, STAG2, PHF6, KMT2C, PPM1D, ATM, ARID1A, ARID2, ASXL2, CHEK2, CREBBP, ETV6, EZH2, FBXW7, MGA, MPL, RB1, SETD2, SUZ12, ZRSR2* or in *CALR* exon 9; (2) any truncating mutations (nonsense, essential splice site or frameshift indel) in known tumor suppressor genes as per the Cancer Gene Census, OncoKB, or the scientific literature; (3) translation start site mutations in *SH2B3*; (4) *TERT* promoter mutations; (5) *FLT3*-ITDs; (6) in-frame indels in *CALR, CEBPA, CHEK2, ETV6, EZH2*; (7) any variant occurring in the COSMIC "haematopoietic and lymphoid" category greater than or equal to 10 times; (8) any variant reported as somatic at least 20 times in COSMIC; (9) any variant noted as potentially oncogenic in an in-house data set of 7000 individuals with myeloid neoplasm greater than or equal to five times; (10) any loci (defined by the amino-acid location) reported as having at least five missense mutations and at least one exact mutational match in TopMed[2].

**Post-processing filters for CH calling**
*MSK cohort.* We applied a series of post-processing filters to further remove false-positive variants caused by sequencing artifacts and putative germline polymorphisms. We removed variants that were found (with a VAF of >2% at least once) in a panel of sequencing data from 300 blood samples obtained from persons under 20 year of age and without evidence of CH. We further filtered single-nucleotide deletions within a homopolymer stretch (≥3 base repetition) of the same deleted base pair, single-nucleotide substitutions completing a stretch of a ≥5-base-pair-long homopolymer (for example, GGCGG→GGGGG), in-frame deletions, or insertions in a highly repetitive region (DUST algorithm score of ≥5) and variants with unequal proportions of forward/reverse direction supporting reads based on a Fisher test. We performed a manual review in Integrative Genomics Viewer of recurrent mutations not previously reported in public databases. We required a VAF of at least 2% and at least ten supporting reads. All genotypes were calculated using sequencing reads and bases with a quality value of at least 20. Because somatic mutations in the blood would be expected to be detected in the blood but not in other tissue compartments, we compared the VAF of mutations in the blood compared with the matched tumor. Variant calls that were present in the blood with a VAF of at least twice that in the tumor, or 1.5 times the VAF if the tumor biopsy site was a lymph node, were considered somatic. This ratio was chosen based on maximizing the sensitivity and specificity of CH calls through simulations of leukocyte contamination in the tumor (see Bolton et al., *Nature Genetics* 2020 for more details). To further filter putative germline polymorphisms that passed the blood/tumor solid tissue ratio due to allelic imbalance in the tumor specimen, we removed any variant reported in any population in the gnomAD database at a frequency >0.005.

*KoCH cohort.* Technically, the requirements to be called positive SNVs/insertions or deletions all sequencing reads have a base-quality value of at least 20 and total read numbers ≥10, Alt read numbers ≥10, positive Alt read numbers ≥5, negative Alt read numbers ≥5, and VAF between 2% and 30%. We further filtered tri-allelic sites and common germline variants with MAF ≥2% in gnomAD (genome aggregation database v.2.1.1), the 1000 Genomes Project release 3, ESP6500 (Exome Sequencing Project v. 6500), the ExAC (Exome Aggregation Consortium) data. At last, to remove technical artifacts 1000 healthy individuals between age 40 and 49 year blood samples were sequenced at depth of 1000×, and CH variant calls were made with the same CH calling pipeline at VAF ≥ 1%. Resulting variants with MAF ≥ 2% and not present in COSMIC database were considered likely artifacts and were filtered.

**Statistical analysis**
*CH and Covid-19 severity.* We used multivariable logistic regression to evaluate for an association between CH and Covid-19 severity adjusting for age (measured as a continuous variable), gender, race, smoking history, and co-existing medical comorbidities including diabetes, COPD/asthma, and cardiovascular disease all classified as per Table 1. This was done separately for the MSK and KoCH cohorts. For solid tumor patients at MSK we also adjusted for primary tumor site (thoracic or non-thoracic cancer) and receipt of cytotoxic chemotherapy before and after IMPACT blood draw. We also performed a sensitivity analysis adjusting for BMI and the month of Covid-19 diagnosis. The OR estimates were largely unchanged; in the model adjusted for BMI and month of Covid-19 diagnosis, this was 1.95 (95% CI 1.17–3.26, p = 0.01). A sensitivity analysis in the MSK-IMPACT cohort limited to those who were not exposed to cytotoxic therapy prior to blood draw yielded similar results (OR = 1.45; 95% CI 0.74–2.84) to those overall for the MSK-IMPACT cohort. Graphical inspection of the frequency of CH by Covid-19 severity stratified by age group and tumor type (for the MSK cohort) suggested that the enrichment of CH among those with severe Covid-19 was not driven by a single age group or tumor type (Supplementary Figs. 5–6).

Given the evidence of similarity in predictors of Covid-19 severity in cancer and non-cancer populations[13,19–21], we hypothesized that the effect of CH on Covid-19 might be comparable between cancer and non-cancer populations. We performed a fixed-effects meta-analysis (using the inverse-variance method in the R package "Metafor"[26]) of the MSK and KoCH cohorts to jointly estimate the odds ratio for severe Covid-19 among CH-positive compared with CH-negative individuals.

*CH and risk of infection in the MSK cohort.* We analyzed billing codes from 14,211 solid tumor patients at MSKCC who had their blood sequenced using MSK-IMPACT. We applied the phecode nomenclature developed at Vanderbilt[6] to map ICD-9 and ICD-10 billing codes to infectious disease subtypes. Subjects who were billed using an ICD-9/10 code within the phecode for the first time following their sequencing blood draw with evidence of CH were considered to have an incident infection. Those who were billed for an ICD-9/10 code within the phecode prior to the blood draw were removed from the analysis of that phecode. In order to evaluate the accuracy of the billing code data, the presence of a documented *C. Difficile* or *Streptococcus* infection in an EMR physician note was manually checked for patients respectively identified by billing codes (N = 525 patients) by three independent physicians using shared criteria for infection onset. Billing codes were highly accurate in identifying the presence of the respective infectious disease (concordance >95%).

We used Cox proportional hazards regression to estimate the HR for risk of infection among those with CH compared with CH-negative individuals. The date of blood draw (used for MSK-IMPACT sequencing) served as the onset date for this time-to-event analysis; the end date was the date of billing code entry for the infectious disease subtype phecode, death, or last follow-up, whichever came first. All models were adjusted for age, gender, race, smoking, tumor type, and cumulative exposure to cytotoxic chemotherapy prior to the blood draw and after blood draw as previously described[10]. Following the 10:1 rule regarding the number of covariates in a multivariable model in proportion to the number of events[16], we excluded infection subclasses populated with <80 individuals. The analysis utilized multiplicity correction with the Benjamini–Hochberg method to establish adjusted q values for HR with a prespecified FDR <0.10. Given the strong relationship between prior antibiotic use and *C. Difficile* infection risk, we performed a sensitivity analysis including the number of courses of antibiotics received after MSK-IMPACT blood draw. There was no clear association between CH and antibiotic use and no effect of inclusion of this as a covariate in the model (Supplementary Fig. 5). Having a history of foley catheter (FC) and/or central venous catheter (CVC) placement was common (37% and 20%, respectively) and more frequent in those who also had a history of streptococcus/enterococcus infection (48% and 34%, respectively). However, similar frequencies of FC and CVC were observed among those with CH (36% and 19%) and without CH (38% and 21%, respectively). Inclusion of a history of FC and CVC placement in the Cox model for risk of streptococcus/enterococcus infection did not impact the HR for CH (HR = 1.58, 95% CI 1.15–2.17, p = 0.004).

All the statistical analyses were performed with the use of the R statistical package (www.r-project.org).

**Reporting summary**. Further information on research design is available in the Nature Research Reporting Summary linked to this article.

## Data availability

All results derived from analysis of clinical sequencing data (CH mutations) for all patients as well as the clinical data (for both the MSK-IMPACT and KoCH cohort) necessary to replicate the findings in the article are available within the Article, Supplementary Data, and on Github: https://github.com/kbolton-lab/papers/tree/main/CH_COVID_NatureComm2021 and Zenodo (https://doi.org/10.5281/zenodo.5293522). The raw sequencing data for the MSK-IMPACT and KoCH cohorts are protected and are not broadly available due to privacy laws. Raw data elements may be requested from zehira@mskcc.org (for MSK-IMPACT) and go01@snu.ac.kr (for KoCH) with appropriate institutional approvals.

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

## Acknowledgements

This work was supported by the National Institute of Health (K08CA241318 to K.L.B, P50 CA172012 to L.B., T32-CA009207 to J.J.), American Society of Hematology (K.L.B.), EvansMDS Foundation (K.L.B.), European Hematology Association (E.P.), Gabrielle's Angels Foundation (E.P.), V Foundation (E.P.), Geoffrey Beene Foundation (E.P), Starr Cancer Consortium (to R.L., A.Z., M.B, R.P.), and the Cancer Colorectal Cancer Dream Team Translational Research Grant (SU2C-AACR-DT22-17 to L.D.). E.P. is a Josie Robertson Investigator. M.M. is supported by funds from the Intramural Research Program of the National Cancer Institute, National Institutes of Health. The KoCH cohort was supported through a grant from the Korea Health Technology R&D Project through the Korea Health Industry Development Institute (KHIDI), funded by the Ministry of Health & Welfare, Republic of Korea (grant number: HI14C1277). We thank the Global Science experimental Data hub Center (GSDC) and Korea Research Environment Open NETwork (KREONET) service for data computing and network provided by the Korea Institute of Science and Technology Information (KISTI). Work performed at Memorial Sloan Kettering Cancer Center was supported in part by the Cancer Center Support Grant (grant no. P30 CA008748), the Marie Josee and Henry R Kravis Center for Molecular Oncology, Cycle for Survival, and MSK Molecular Diagnostics Service.

## Author contributions

K.L.B., M.B.F., J.Jee, A.S., Y.K., H.I., C.H.S., L.A.D., L.N., R.L.L., P.N., M.J.M., P.A., E.S.K., N.J.K., A.Z conceived and designed the study. K.L.B., M.B.F., J.Jee, K.A.T., C.H.S., A.S., J.H.M., J.Y.L., J.J., C.K.K., K.S., P.G.C., W.B.P., L.Z.B., H.B.K., M.O., H.S., S.K., M.P., E.S.K., N.J.K. performed collection of clinical data. K.L.B., H.I., R.P., E.G., A.S., T.G., E.P., M.L., M.F.B., C.H.S., A.Z led the generation and analysis of sequencing data. K.L.B., M.B.F., K.O., V.J., J.Jee., A.S., A.D., N.E.B., M.S.P., M.K., Y.K., H.I., B.J.W., I.C.C., M.J.R., C.H.S., S.K., H.S., A.Z., performed statistical analyses and/or participated in data interpretation. All authors contributed to the writing of the manuscript and approved it for submission.

## Competing interests

The authors declare the following competing interests: K.B. has received research funding from GRAIL and Bristol Myers Squibb; Y.K. is a co-founder in Genome Opinion. M.F.B. is on the advisory board for Roche and receives research support from Illumina. J.J. has a patent licensed by the company MDSeq Inc. R.L.L. is on the supervisory board of Qiagen and is a scientific advisor to Loxo, Imago, C4 Therapeutics, and Isoplexis, which include equity interest. He receives research support from and consulted for Celgene and Roche and has consulted for Lilly, Janssen, Astellas, Morphosys, and Novartis. He has received honoraria from Roche, Lilly, and Amgen for invited lectures and from Gilead for grant reviews. A.Z. received honoraria from Illumina. E.P. receives research funding from Celgene. D.G. and has received honoraria for speaking and scientific advisory engagements with Celgene, Prime Oncology, Novartis, Illumina, and Kyowa Hakko Kirin and is a co-founder in Isabl Technologies. M. Ladanyi has served on the advisory boards for AstraZeneca, Bristol Myers Squibb, Takeda, Bayer, BluePrint, Pfizer, Janssen, and Merck, and has received research support from Loxo Oncology and Helsinn Therapeutics, and Elevation Oncology. L.A.D. is a member of the board of directors of Personal Genome Diagnostics (PGDx) and Jounce Therapeutics; is a paid consultant to PGDx and Neophore; is an uncompensated consultant for Merck (with the exception of travel and research support for clinical trials); is an inventor of multiple licensed patents related to technology for circulating tumor DNA analyses and mismatch repair deficiency for diagnosis and therapy from Johns Hopkins University, some of which are associated with equity or royalty payments directly to Johns Hopkins and L.A.D.; and holds equity in PGDx, Jounce Therapeutics, Thrive Earlier Detection and Neophore; his wife holds equity in Amgen. The terms of all of these arrangements are being managed by Johns Hopkins and Memorial Sloan Kettering in accordance with their conflict of interest policies. H.I., C.H.S., H.S., S.K. are current employees of Genome Opinion and holds stock in the company. All other authors declare no competing interests.

## Additional information

[1]Department of Medicine, Washington University, St Louis, MO, USA. [2]Department of Internal Medicine, Seoul National University Hospital, Seoul, Korea. [3]Genome Opinion Inc., Seoul, Korea. [4]Center for Precision Medicine, Seoul National University Hospital, Seoul, Korea. [5]Department of Medicine, Memorial Sloan Kettering Cancer Center, New York, NY, USA. [6]Department of Pathology, Memorial Sloan Kettering Cancer Center, New York, NY, USA. [7]Department of Internal Medicine, Kyungpook National University Hospital, School of Medicine, Kyungpook National University, Daegu, Korea. [8]Department of Internal Medicine, National Medical Center, Seoul, Korea. [9]Department of Internal Medicine, Seoul National University Bundang Hospital, Seongnam, Korea. [10]Center for Hematologic Malignancies, Memorial Sloan Kettering Cancer Center, New York, NY, USA. [11]Department of Epidemiology & Biostatistics, Memorial Sloan Kettering Cancer Center, New York, NY, USA. [12]Department of Radiation Oncology, Memorial Sloan Kettering Cancer Center, New York, NY, USA. [13]Computational Oncology Service, Department of Epidemiology & Biostatistics, Center for Computational Oncology, Memorial Sloan Kettering Cancer Center, New York, NY, USA. [14]Department of Laboratory Medicine, Memorial Sloan Kettering Cancer Center, New York, NY, USA. [15]Department of Pathology and Molecular Medicine, Queen's University, Kingston, ON, Canada. [16]Cardiovascular Research Center, Massachusetts General Hospital, Boston, MA, USA. [17]Department of Medicine, Brigham and Women's Hospital, Harvard Medical School, Boston, MA, USA. [18]Division of Cancer Epidemiology and Genetics, National Cancer Institute, Bethesda, MD, USA. [19]Ontario Institute for Cancer Research, Toronto, ON, Canada. [20]Clinical Genetics Research Lab, Department of Medicine, Memorial Sloan Kettering Cancer Center, New York, NY, USA. [21]These authors contributed equally: Kelly L. Bolton, Youngil Koh, Michael B. Foote, Hogune Im, Justin Jee, Choong Hyun Sun, Anton Safonov, Eu Suk Kim, Nam Joong Kim, Ahmet Zehir.
✉email: bolton@wustl.edu; eskim@snubh.org; molder@unitel.co.kr; zehira@mskcc.org

