## [Peer Review File · Nature Communications]

Clonal hematopoiesis is associated with risk of severe Covid-19Editorial Note: This manuscript has been previously reviewed at another journal that is not operating a transparent peer review scheme. This document only contains reviewer comments and rebuttal letters for versions considered at *Nature Communications*.

REVIEWER COMMENTS

Reviewer #1 (Remarks to the Author):

I thank the authors for their responses, which address most of my comments.

I have one remaining concern which pertains to my “overall comment” as Reviewer #1 that was unfortunately, misunderstood. so I will try to be clearer:

An important limitation of this manuscript is that what most readers will understand as “clonal hematopoiesis”, i.e. clonal hematopoiesis with putative drivers (CH-PD), was not associated with severe COVID-19 disease. Instead non-PD CH was associated with severe COVID-19 disease.

Non-PD CH is not well understood and this is particularly so when/if it arises after chemotherapy. My comment was intended to help formulate a hypothesis for the association with non-PD CH, which can be (partially) tested. I ask the authors to consider whether non-PD CH arises in individuals with a poor hematopoietic reserve after chemotherapy and is a reflection of higher frailty/poor fitness (i.e. similarly to PPM1D-CH, which arises commonly after chemotherapy and is associated with poor survival outcomes, and appears enriched in those with severe COVID-19 in this study).

In fact, when the authors focused their analysis on those not exposed to cytotoxic chemotherapy, they found a lower & non-significant OR for severe COVID-19 (1,45; 95% CI 0.74-2.84) vs the group overall (1.85; 95% CI 1.15-2.99). This is in keeping with the above hypothesis (with the caveats of sample size etc). It would help if the authors also performed a focused analysis on those exposed to chemotherapy and discuss their findings appropriately.

Also, as PD-CH was not associated with severe COVID-19, it would be prudent to mention this in the abstract.

Reviewer #1 (Remarks to the Author):

I thank the authors for their responses, which address most of my comments.

1) I have one remaining concern which pertains to my “overall comment” as Reviewer #1 that was unfortunately, misunderstood. so I will try to be clearer:

An important limitation of this manuscript is that what most readers will understand as “clonal hematopoiesis”, i.e. clonal hematopoiesis with putative drivers (CH-PD), was not associated with severe COVID-19 disease. Instead non-PD CH was associated with severe COVID-19 disease.

Non-PD CH is not well understood and this is particularly so when/if it arises after chemotherapy. My comment was intended to help formulate a hypothesis for the association with non-PD CH, which can be (partially) tested. I ask the authors to consider whether non-PD CH arises in individuals with a poor hematopoietic reserve after chemotherapy and is a reflection of higher frailty/poor fitness (i.e. similarly to PPM1D-CH, which arises commonly after chemotherapy and is associated with poor survival outcomes, and appears enriched in those with severe COVID-19 in this study).

In fact, when the authors focused their analysis on those not exposed to cytotoxic chemotherapy, they found a lower & non-significant OR for severe COVID-19 (1.45; 95% CI 0.74-2.84) vs the group overall (1.85; 95% CI 1.15-2.99). This is in keeping with the above hypothesis (with the caveats of sample size etc). It would help if the authors also performed a focused analysis on those exposed to chemotherapy and discuss their findings appropriately.

We thank Reviewer #1 for explaining this further and in response to this comment have performed additional analyses on the relationship between CH and COVID-19 stratified by exposure to chemotherapy. We also have performed additional analyses specifically on the relationship between chemotherapy and non-PD CH.

In the full cohort of MSK-IMPACT patients (N=21,136), we studied the relationship between non-PD CH and demographic characteristics including prior exposure to cytotoxic therapy. As for CH-PD, non-PD CH was strongly associated with Age ($p=4 \times 10^{-77}$). However, unlike CH-PD, non-PD CH was not associated with prior exposure to cytotoxic therapy (OR=1.0, 95% CI 0.92-1.08, $p=0.92$). The full model is shown at the end of this document. Because tumor type in many ways accounts for therapy patterns, we re-ran the model but excluding tumor type. This yielded similar results (OR=0.99, 95% CI 0.92-1.08, $p=0.9$).

We studied the relationship between non-PD CH specifically and COVID-19 severity stratified by prior exposure to cytotoxic therapy specifically in the MSK-IMPACT cohort. We observed a similar effect in both individuals who were exposed to cytotoxic and those who were unexposed to cytotoxic therapy (taking into account the 95% confidence intervals to guide our interpretation). For those with no prior exposure to cytotoxic therapy, the OR was 1.88, 95% CI 0.98-3.90, $p=0.09$. For those with prior exposure to cytotoxic therapy the OR was 2.12, 95% CI 0.71-6.22, $p=0.17$. Thus, our conclusion is that the relationship between non-PD CH and COVID-19 severity does not appear to be greatly modified by exposure to cytotoxic therapy.

Separately, we studied the relationship between CH-PD and exposure to COVID-19 severity stratified by exposure to therapy. Similar to our findings overall for CH-PD, among those not

previously exposed to chemotherapy, no association was observed between CH-PD and COVID-19 severity (OR=0.82, 95% CI 0.34-1.83, p=0.63). We did however, observe a non-significant trend towards a positive association between CH-PD and COVID-19 severity among those previously exposed to chemotherapy (OR=2.4, 95% CI 0.86-6.74, p=0.09). As the reviewer points out we also observed a trend towards a higher proportion of individuals with CH-PD mutations in *PPM1D* among those with severe COVID-19 (5%, N=5) compared to non-severe COVID-19 (1%, N=2) (Supplementary Figure 2). This provides some evidence that CH-PD in cancer patients with severe COVID-19 may reflect poor hematopoietic reserve after chemotherapy and in turn worse fitness. This would need to be further studied in larger populations of cancer patients but is an interesting observation.

We have modified the text to include these updated analyses but emphasize the preliminary nature of these findings. This begins on page 6 line 25:

“To study whether exposure to chemotherapy might modify the relationship between non-PD CH, PD-CH and COVID-19 severity, we performed a series of stratified analyses by prior exposure to cytotoxic chemotherapy within the MSK cohort. We observed a similar enrichment of non-PD CH among individuals with severe COVID-19 among those exposed to cytotoxic therapy (2.12, 95% CI 0.71-6.22, p=0.17) and unexposed (1.88, 95% CI 0.98-3.90, p=0.09). The relationship between non-PD CH and COVID-19 severity did not appear to be greatly modified by exposure to cytotoxic therapy. Similar to our findings overall for CH-PD, among those not previously exposed to chemotherapy, no association was observed between CH-PD and COVID-19 severity (OR=0.82, 95% CI 0.34-1.83, p=0.63). We did however, observe a non-significant trend towards a positive association between CH-PD and COVID-19 severity among those previously exposed to chemotherapy (OR=2.4, 95% CI 0.86-6.74, p=0.09). Of note, we also observed a trend towards a higher proportion of individuals with CH-PD mutations in *PPM1D* among those with severe COVID-19 (5%, N=5) compared to non-severe COVID-19 (1%, N=2) (Supplementary Figure 2). This provides some evidence that CH-PD in cancer patients with severe COVID-19 may reflect poor hematopoietic reserve after chemotherapy and in turn worse immune response to infection. However, this would require further investigation in larger populations of cancer patients.”

2) Also, as PD-CH was not associated with severe COVID-19, it would be prudent to mention this in the abstract.

We agree this would be helpful to mention in the abstract and have updated the abstract as follows:

Acquired somatic mutations in hematopoietic stem and progenitor cells (clonal hematopoiesis or CH) are associated with advanced age, increased risk of cardiovascular and malignant diseases, and decreased overall survival.¹⁻⁴ These adverse sequelae may be mediated by altered inflammatory profiles observed in patients with CH.^{2,5,6} A pro-inflammatory immunologic profile is also associated with worse outcomes of certain infections, including SARS-CoV-2 and its associated disease Covid-19.^{7,8} Whether CH predisposes to severe Covid-19 or other infections is unknown. Among 525 individuals with Covid-19 from Memorial Sloan Kettering (MSK) and the Korean Clonal Hematopoiesis (KoCH) consortia, we found that CH was associated with severe Covid-19 outcomes (OR=1.85, 95%=1.15-2.99, p=0.01), **in particular CH characterized by**

non-cancer driver mutations (OR=2.01, 95% CI=1.15-3.50, p=0.01). We further explored the relationship between CH and risk of other infections in 14,211 solid tumor patients at MSK. CH was significantly associated with risk of *Clostridium Difficile* (HR=2.01, 95% CI: 1.22-3.30, p=6x10⁻³) and *Streptococcus/Enterococcus* infections (HR=1.56, 95% CI=1.15-2.13, p=5x10⁻³). These findings suggest a relationship between CH and risk of severe infections that warrants further investigation.

Logistic regression for non-PD CH				
Term	OR	95% conf.low	95% conf.high	p.value
Age	1.65	1.57	1.74	4.45E-77
Smoking status (ref=never)				
current_former	1.17	1.08	1.28	1.70E-04
missing	1.60	0.91	2.67	8.42E-02
Race				
ASIAN-FAR EAST/INDIAN SUBCONT	0.94	0.80	1.10	0.47
BLACK OR AFRICAN AMERICAN	1.07	0.91	1.25	0.42
HISPANIC OR LATINO	0.93	0.78	1.11	0.45
OTHER_UNKNOWN	0.87	0.71	1.05	0.16
Male Gender	0.93	0.84	1.02	0.11
Cytotoxic Therapy (prior to IMPACT)	1.00	0.92	1.08	0.92
Tumor Type (ref=lung)				
Breast	1.02	0.79	1.32	0.88
CNS	1.13	0.84	1.52	0.41
gastrointestinal	1.00	0.79	1.27	1.00
genitourinary	0.98	0.75	1.29	0.90
gynecologic	1.04	0.80	1.34	0.79
head_neck	1.09	0.78	1.50	0.62
skin	0.97	0.70	1.35	0.85
thoracic	1.05	0.83	1.35	0.67
unknown_primary	1.09	0.80	1.50	0.59

REVIEWER COMMENTS

Reviewer #1 (Remarks to the Author):

I thank the authors for adequately addressing my comments.